

# Impacts of Barrier-Island Breaching on Mainland Flooding During Storm Events applied to Moriches, NY

Catherine R. Jeffries[1], Robert Weiss[2], Jennifer L. Irish[3], and Kyle Mandli[4]

[1]Department of Geosciences & Center for Coastal Studies, Virginia Tech, Blacksburg, VA
[2]Academy of Integrated Science & Department of Geosciences & Center for Coastal Studies, Virginia Tech, Blacksburg, VA
[3]Department of Civil and Environmental Engineering & Center for Coastal Studies, Virginia Tech, Blacksburg, VA
[4]Department of Applied Physics and Applied Mathematics, Columbia University, New York City, NY

**Correspondence:** Catherine R. Jeffries (catherinej@vt.edu)

**Abstract.** Barrier islands can protect the mainland from flooding during storms through reduction of storm surge and dissipation of storm generated wave energy. However, the protective capability is reduced when barrier islands breach and a direct hydrodynamic connection between the water bodies on both sides of the barrier island is established. Breaching of barrier islands during large storm events is complicated, involving nonlinear processes that connect water, sediment transport, dune height, and island width among other factors. In order to assess how barrier island breaching impacts flooding on the mainland, we used a statistical approach to analyze the sensitivity of mainland storm-surge to barrier island breaching by randomizing the location, time, and extent of a breach event. We created a framework that allows breaching to develop during the course of a simulation and imposes a breach in an approximation of a Gaussian bell-curve that deepens over time. We show that simulating a storm event and varying the size, location, and number of breaches in the barrier island that mainland storm surge and horizontal inundation is affected by breaching; total inundation has a logarithmic relationship with total breach area which tapers off after the entire island is removed. Breach location is also an important predictor of inundation and bay surge. The insights we've gleaned from this study can help prepare shoreline communities for the differing ways that breaching affects the mainland coastline. Understanding which mainland locations are vulnerable to breaching, planners and coastal engineers can design interventions to reduce the likelihood of a breach occurring in areas adjacent to high flood risk.

## 1 Introduction

Barrier islands are long, shore-parallel, low-relief land masses that are adjacent to approximately 6.5% of the world's coastlines (Oertel, 1985; Stutz and Pilkey, 2001). According to Oertel (1985) barrier island systems consist of six sedimentary environments; proximity to the mainland, a back-barrier region (bay or lagoon), an inlet and inlet delta, the barrier island, the barrier platform, and the shoreface. Barrier islands are protective structures that help dissipate wave energy and storm surge approaching the mainland from the ocean. Barrier island dunes that are higher than the approaching storm surge cause wave breaking, which reduces the impact of the surge when it reaches the back-barrier bay (Oertel, 1985; Irish et al., 2010). Vegetated low-lying dunes are more resistant to erosion and will absorb some of the seaward driven surge and wave energy. The dissipation of wave energy ensures that barrier islands undergo significant changes during storms and hurricanes, one of which is breaching.



A breach is an opening in a narrow landmass, such as a barrier island, that allows a direct hydrodynamic connection between the ocean and the back-barrier bay or lagoon (Kraus, 2003; Wamsley and Kraus, 2005). Naturally occurring breaching is a complex process that involves the interaction of storm surge, waves and their resulting overwash with barrier island width and height. Storm forcing combined with local bathymetry is necessary to initiate the conditions that lead to breaching. Variations in storm size, intensity and locale may cause breaches in some locations but not in others.

Large storms, such as hurricanes, can devastate barrier islands and the mainland coastline. One of the many hazards presented by such storms is storm surge, an abnormal rise in sea level driven by wind and atmospheric pressure changes. Storm surge that causes a water level gradient between the ocean and back-barrier region forces water to flow rapidly over the barrier island, eroding sediment to equalize the water level. This gradient involves a critical elevation of water levels that may not inundate the island, but can still cause erosion (Kraus et al., 2002; Kraus, 2003). Storm surge and wave setup increase water elevation in the ocean and the back-barrier region; these water levels along with wave action, reduce the stability of the barrier island dune slope (Kraus, 2003; Kraus et al., 2002). However, wave attack alone, while weakening the dune slope, is unlikely to induce breaching because the net erosion is seaward and does not push erosion landward (Pierce, 1970).

During storm-induced overwash and inundation, water flowing across the island can scour a channel between the sea and the back-barrier region (Kraus, 2003; Pierce, 1970; Roelvink et al., 2009). This scouring requires strong flow and sustained inundation. Breaching can occur from both the seaward and landward side of the barrier island, but field data are limited in showing the direction of breach initiation (Kraus, 2003; Pierce, 1970; Smallegan and Irish, 2017). However, (Smallegan and Irish, 2017) show that bay surge following peak ocean surge is more likely to cause breaching from the landward side, as peak ocean surge already weakened the dune through erosion caused by wave attack and swash (Kraus, 2003; Smallegan and Irish, 2017). Identifying breach locations is challenging; localized lows for dune height and narrower portions of the island are more likely potential breach locations (Kraus, 2003; Kraus and Wamsley, 2003). One study by (van der Lugt et al., 2019) simulated Hurricane Sandy (2012) using surge-tide levels, 2D wave-spectra, and sediment transport to model barrier island morphodynamics with pre-storm LiDAR bathymetric grids. The sediment transport model generated two breaches at locations at peak erosion sites, but neither breach location matched the observed breach that opened during Hurricane Sandy (van der Lugt et al., 2019).

Quantifying breach dimensions during hurricanes is challenging. While breach growth over time has been documented (Kraus and Wamsley, 2003; Schmeltz et al., 1982),these studies focus on days to months post-storm. Predicting breach locations and tracking their growth during a hurricane is not feasible. Lab and field experiments by (Visser, 1999) for breaches in dikes are useful but the breach is initiated with a pre-drilled hole in the dike and does not simulate exactly what occurs to barrier islands during storms. (Buynevich and Donnelly, 2006) performed a geologic mapping of some New England, USA barrier islands and found geologic signatures to indicate the islands' past history with breaching and overwash. (Buynevich and Donnelly, 2006) found ephemeral breaches with widths of 10 - 30 m before closing and breach depths of one - three meters below the dune crest. Some post-storm surveys have defined breach sizes before natural or forced closing. (Kraus and Wamsley, 2003) discusses Pike's Inlet on Long Island, New York, USA which was initially measured after the hurricane at 304.8 m wide and a nearby breach named Little Pike's Inlet was initially 30.48 m wide but over several months grew to over 914.4 m before it was



closed. A breach near Moriches Inlet on Long Island studied by (Schmeltz et al., 1982) had an initial size of 91.4 m and 0.61

60   m depth. This breach expanded to 885 m with a depth of three meters before it was mechanically closed. The uncertainties in

breach dimensions and in where, how, and when breaches occur remains one of the many issues facing coastal communities

today, due to the inability to predict or plan for the probable impacts of a breach forming where populations are highest.

Barrier islands are found along the coasts of 18 US states bordering the Atlantic Ocean and Gulf of Mexico (Zhang and

Leatherman, 2011). As coastal populations have increased significantly in recent decades, the protective nature of barrier

islands has become more crucial (Zhang and Leatherman, 2011). According to the US National Hurricane Center (NHC),

storm surge is the leading cause of loss of life and property damage during hurricanes (National Hurricane Center, 2006).

Storm surge can cause flooding that damages structures, closes roads, and disrupts coastal communities. It can also accelerate

erosion on barrier islands and the mainland, increasing flood risk. Understanding how barrier island breaching affects coastal

flooding from storm surge is vital for risk assessment and mitigation. A hydrodynamic connection between the ocean and

back-barrier region can lead to increased flooding and wave action during hurricanes, heightening risks to populations and

property. However, there is little information on how different breach morphodynamics affect the mainland.

In this paper, we explore the different inundation patterns and surge depths at Moriches, New York, USA for a storm

that approximates the 1938 Hurricane. Using GeoClaw, software capable of modeling storm surge, we artificially alter the

bathymetry of a barrier island during a storm simulation to create breaches in the barrier island (Mandli et al., 2016). This

method removes the complexities of modeling the morphological processes driving breach formation so we can purely study

the coastal response to these breaches. We randomized the number, width, and depth of these synthetic breaches to gain a

statistical understanding of how these parameters influence coastal inundation and bay storm surge.

## 2   Methods

### 2.1   Study Area - 1938 Hurricane

Our study focuses on Moriches, NY a section of the barrier island that spans Long Island, New York, USA along the Atlantic

Ocean. This region is heavily populated and is regularly impacted by storms. It was especially damaged by the 1938 hurricane.

The barrier island that runs roughly parallel to the mainland at Moriches Bay is comprised of two segments of a larger barrier

island system that is adjacent to the southern portion of Long Island, NY. Fire Island, the center island in this system, connects to

Westhampton Beach via Moriches Inlet and is the region we are studying. This segment of the barrier system is approximately

20 km long with a width that varies from 0.25 km to 0.50 km between the ocean and bay and has a dune height ranging from

zero to nine m above mean sea level (MSL) (Leatherman, 1999; Cooperative Institute for Research in Environmental Sciences,

2021).

The 1938 Hurricane made landfall as a category 3 hurricane near Moriches, NY on September 21, 1938. The center of the

storm passed over the western side of Great South Bay, less than 75 km from Moriches Bay. Figure 1 illustrates the location

of Moriches Bay, NY and the storm's location as it made landfall. It generated 10 breaches across the barrier island system





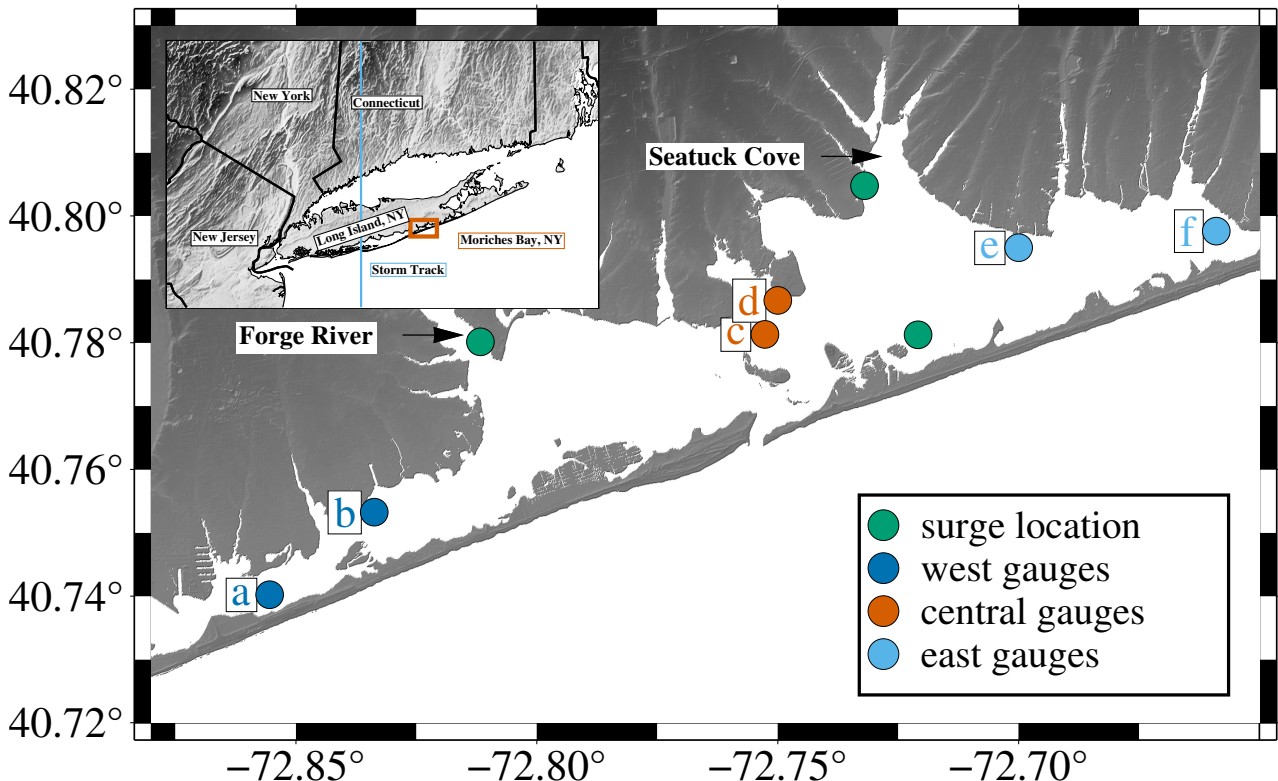

**Figure 1.** Map of study area Moriches, NY. Inset shows region surrounding Moriches NY (orange box). Storm track for the 1938 Hurricane (light blue line) and our simulated storm. Green circles are locations of surge measurements from Table 2. Remaining circles are locations of synthetic water level gauges illustrated in Fig. 4.

and caused widespread damage (Morang, 1999; Coch, 1994; Cañizares and Irish, 2008). Six breaches were opened during the hurricane at Moriches, NY specifically, three each on either side of the inlet.

## 2.2 GeoClaw

Our goal for this project is to quantify the differences in coastal and bay flooding if breaching occurs during a hurricane. To simulate the storm, we used GeoClaw, a software package that solves depth-averaged fluid equations in one and two dimensions to model geophysical events (Clawpack Development Team, 2020; Mandli et al., 2016). GeoClaw employs adaptive mesh refinement (AMR) that allows for increasing resolution where and when it is needed, reducing computational overhead while providing an accurate solution (Berger et al., 2011). GeoClaw has been validated by the US National Tsunami Hazard Mitigation Program (NTHMP) for tsunami modeling. González et al. (2011) describes the benchmarking process used to validate GeoClaw in that case.

Storm surge modeling with GeoClaw has been proposed to provide a robust but less computationally expensive model than ADCIRC, a commonly utilized finite element model (Westerink et al., 2008; Mandli and Dawson, 2014; Bates et al., 2021).



Mandli and Dawson (2014) compared GeoClaw with ADCIRC in a simulation of Hurricane Ike and verified the accuracy of the solution against in situ water level gauges that recorded data during the hurricane. GeoClaw calculates storm surge with a 105 two-dimensional model averaged in depth that solves the shallow water equations with source terms for bathymetry, bottom friction, Coriolis forcing, surface pressure, and wind friction (see Mandli and Dawson (2014) for further details). GeoClaw's default storm surge modeling does not provide tidal, riverine, or wave-stress calculations that are included in other models currently used in practice, such as ADCIRC (Westerink et al., 2008; Mandli and Dawson, 2014). For the purposes of solely studying the impact of breach dimensions and locations on the mainland flooding we did not seek to add tidal, or wave induced 110 surge to our simulations.

## 2.3 Storm Forcing

The storm we used to simulate storm surge is a proxy for the 1938 New England Hurricane. The storm data was generated for the US Army Corps of Engineers (USACE) North Atlantic Comprehensive Coast Survey (NACCS) (Cialone et al., 2015; Nadal-Caraballo et al., 2015). The suite of synthetic tropical cyclones developed for the NACCS study were generated with a 115 planetary bound layer (PBL) model that utilizes six storm parameters: the storm track, the heading direction, the central pressure deficit, the radius of max winds, and the translational speed. Each parameter was sampled from historical tropical cyclones from National Oceanographic and Atmospheric Administration (NOAA) National Hurricane Center HURDAT2 (HURricane DATa 2nd generation) (Landsea and Franklin, 2013) database and then were used to generate probability distributions. The probability distributions provided combinations of parameters that were input into the PBL model, which calculated the wind 120 and pressure data for use in simulations. These data were validated post simulation using several historical hurricanes and given an uncertainty of 0.39 meters for the region covering New York and New Jersey (Nadal-Caraballo et al., 2015).

The storm forcing is provided by wind and pressure fields with data in 15-minute increments and at 0.25 degree spatial resolution. GeoClaw's AMR algorithms require data to be integrated both temporally and spatially over the course of the simulation. This ensures that the solution evolves in space and time as the hurricane progresses (Mandli and Dawson, 2014; 125 Berger et al., 2011). We utilized linear interpolation of the wind and pressure data between time steps calculated by GeoClaw's AMR module (Mandli and Dawson, 2014). To define the wind and pressure forcing inside the AMR grids, we employed bi-linear interpolation when and where required. The chosen storm has a track and intensity similar to that of the 1938 Hurricane. We verified the accuracy of the solution using a water level gauge (Station ID: 8531680) at Sandy Hook, NJ with data recorded from 1938 (Center for Operational Oceanographic Products and Services (CO-OPS), 2007). Figure 2 illustrates our validation 130 of the storm forcing. The original data recorded during the hurricane is referenced to local MSL of the time, with the tidal component removed. We adjusted the data upwards 0.239 m to match the modern datum used by NOAA (Center for Operational Oceanographic Products and Services (CO-OPS), 2007), accounting for sea-level rise at that station. Our synthetic water level gauge data is calculated with bathymetry that uses NAVD88 as the vertical datum, which is 0.073 m above the local MSL for Sandy Hook. We reduced the synthetic gauge data to bring both the synthetic and original data to the same datum. As shown 135 in Fig. 2 the water level from the simulation increases similarly to the recorded data, although the peak occurs 1.30 hours





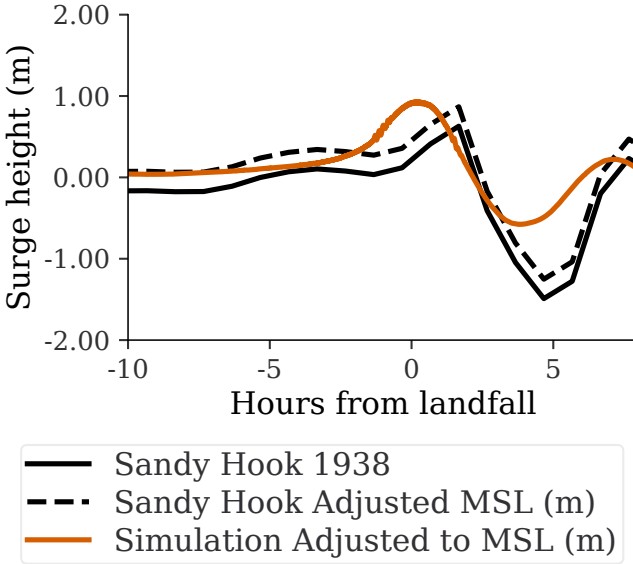

**Figure 2.** NOAA water level gauge data from the 1938 hurricane compared with simulated water level gauge data at Sandy Hook, NJ. Black line is original data before adjusting for modern mean sea levels. Dotted black lines are the data adjusted 0.239 to match the datum used by NOAA. Orange line is the simulation output adjusted -0.073 for the difference between NAVD88 and local MSL.

earlier, with a 0.05-meter difference from the recorded data. However, given the changes in local bathymetry since 1938, we feel confident this is a reasonable approximation of the 1938 Hurricane.

The storm approached Long Island, NY directly from the south. The track is illustrated with a blue line on the inset of Fig. 1. As the storm approached, storm surge seaward of the barrier island began to rise from southwest to northeast starting

approximately 18 hours before landfall. The timing of the peak ocean surge varies along the barrier island, with the peak occurring in the northeast approximately ten minutes after the southwest, which occurred within minutes of landfall. Inside the bay, a local wind setup occurred from northeast to southwest before landfall. After peak surge, water was pushed into the southwest of the bay from Great South Bay. A small setdown then occurred three hours after landfall as the storm continued north.

**2.4   Breaching**

Due to breaching's complex nature and the lack of studies that define breaching during hurricanes, we applied a simple framework to reduce dune height at specified locations that take the shape of a gaussian function. During the storm simulation we apply equation 1 to reduce the height of the barrier island at a selected location, where $d^t$ is the height of the breach location at time $t$, $X$ is the alongshore coordinate of the location being reduced $\mu$ is the center of the breach location, and $t_T$ is a timing

factor that controls how quickly the breach opens.



$$d^t = d^{t-1} - e^{-\frac{1}{2}(X-\mu)^2} t_T \tag{1}$$

The timing factor for these simulations allows for the breach to open fully in an hour after the experiments performed by Visser (1999). The calculation to reduce the dune height is controlled in the cross shore by the latitudes that are the transition between land and water for the bay and ocean. Figure 3 is a schematic to illustrate equation 1.

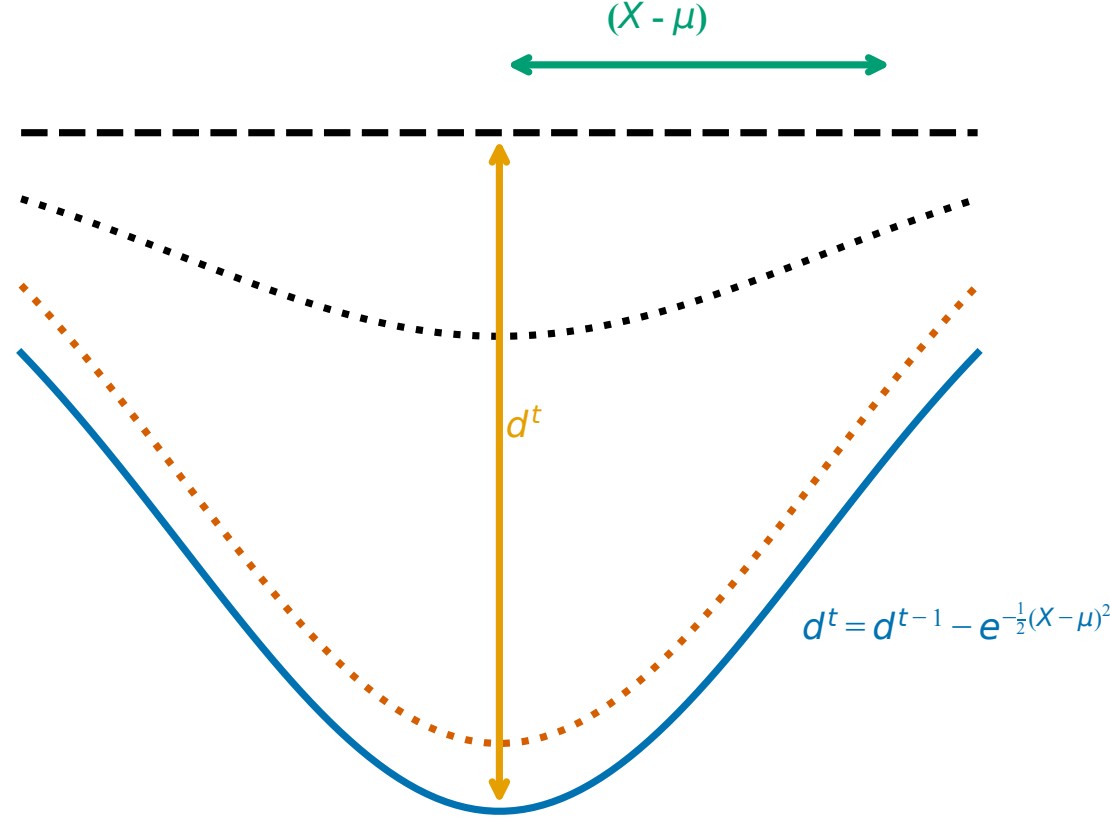

**Figure 3.** Schematic of breach growth based on equation 1 ($X$ - $\mu$) is width of breach for each location lowered. $d^t$ is total depth of center of breach. Black dashed line indicates original barrier height. Black dotted line indicates breach growth at an intermediate time $t$. Orange dotted line is breach growth at nearly final time $t$. Blue line is final breach.



## 2.5 Simulations

We used the GEBCO 30 arc second (Weatherall et al., 2015) bathymetry for the region covering 98W to 57W and 5N to 45N with a maximum resolution of 6750 meters. For Moriches, NY we also used NOAA's continuously updated 1/9 arc-second topobathy dataset (Cooperative Institute for Research in Environmental Sciences, 2021). We restricted GeoClaw's adaptive refinement for Moriches Bay and its adjacent barrier island to an 18 x 18 meter grid, balancing resolution with compute time. The refinement begins well before the storm arrives to more accurately simulate the surge as it enters the bay. We placed synthetic water level gauges in the bay using the NACCS project maximum surge locations (Cialone et al., 2015) with additional gauges on the seaward side of the barrier every two km to verify nearshore surge height for breach initiation.

To create a baseline for quantifying the flooding and inundation changes in different simulations, we ran a no-breach scenario. This was necessary to observe how the storm impacts the barrier island, bay, and coastline in the absence of breaching. The water level gauges we placed in the bay and on the seaward side of the barrier island help discern differences in surge height and timing compared to the other simulations. A challenge of this study is that the barrier island at Moriches, NY has undergone significant topographic and morphological changes since 1938. To ensure that the original breaches would open under these changed conditions we decided that a water level gauge near the seaward side of the island must record a water level that is a percentage of the modern dune height. The highest offshore water level recorded for all six breach locations was 24% of the dune height, which we used as a criterion for identifying sites that could be breached throughout the barrier island.

Our simulations vary the width, depth, location, and number of breaches. For the first group of scenarios, we used the original breach locations formed during the 1938 Hurricane. We estimated the original breach dimensions from observations reported in Cañizares and Irish (2008) and references therein. The specified locations used a single longitude as a center point ($\mu$) from Equation 1. We used the latitudes that touch the bay and ocean directly north and south from $\mu$. We created a Monte-Carlo framework that employs a random, uniform distribution for each breach's width and depth. The initiation time for each breach was chosen using the no-breach simulation results for the nearest synthetic water level gauge seaward of the island. The first time the nearest water level gauge reached 24% of the maximum dune height at each location was set as the breach initiation time. We chose to have the breaches fully open within one hour after initiation.

We created six broad categories of simulations as shown in Table 1, *Width*, *Depth*, *Width and Depth*, *Locations*, *East of Inlet*, and *West of Inlet*. In *Width* we randomized the width of the breaches, held the depth at two m below MSL and used the original six breach locations from the 1938 Hurricane. Each breach used boundary values between 25 meters and 630 meters, reflecting the few details on initial breach size found in the literature (Schmeltz et al., 1982; Kraus and Wamsley, 2003; Visser, 1999; Cañizares and Irish, 2008). For the *Depth* simulations we kept the 1938 breach width estimates and varied depths between zero and two m below MSL. In *Width and Depth* we randomized both dimensions and varied the number of breaches from one to five, with each breach at one of the original locations randomly chosen. The *Location*, *East of Inlet*, and *West of Inlet* categories varied all breach parameters, including the number, dimensions, and locations of breaches. Our Monte-Carlo framework selected alongshore coordinates from a full list for the *Location* category, while for the *East* and *West* categories, we restricted breach locations to sections on either side of the inlet. After selecting the alongshore coordinate our algorithm



**Table 1.** Breach input data for the 1900 simulations for our study. Each category of simulation's widths and depths were chosen by Monte-Carlo methods using a random uniform distribution between endpoints identified from literature

|  | Num. Breaches | Min. Width | Max. Width | Min. Depth | Max. Depth |
|---|---|---|---|---|---|
| Width Simulations | 6 | 25.00 | 628.97 | -2.00 | -2.00 |
| Depth Simulations | 6 | 110.40 | 627.90 | -2.02 | -0.00 |
| Width and Depth Simulations | 1 - 5 | 25.00 | 628.97 | -2.02 | -0.00 |
| Locations | 1 - 294 | 25.00 | 629.00 | -2.01 | -0.00 |
| East of Inlet | 1 - 99 | 25.00 | 629.00 | -2.01 | -0.00 |
| West of Inlet | 1 - 99 | 25.00 | 628.99 | -2.01 | -0.00 |

verified that the maximum surge height at the nearest ocean gauge reached 24% of the dune height. If no gauges within two
kilometers met that threshold, a new location was chosen. This ensures a reasonable estimate of breach-inducing conditions,
as locations where the offshore water levels do not reach that critical elevation are unlikely to breach. We constrained the
maximum number of breaches using a nearshore surge height of 24% of the dune height, identifying up to 295 viable breach
locations along the island. This high number compared to the six original breaches from 1938 underscores the non-linearity
and stochastic nature of breaching processes. For the *East* and *West* simulations, we limited breaches to fewer than 100 per
section.

## 2.6 Data Analysis

We evaluated our results using the total horizontal inundation along the mainland coastline and the maximum surge height data
recorded by GeoClaw for the entire bay, at specific points within the bay (green circles in Figure 1), and the surge time-series
generated at select synthetic water level gauges (blue and orange circles on Fig. 1).

### 2.6.1 Water Level Gauges

GeoClaw records time series data from synthetic water level gauges at five minute intervals during the storm simulation. We
gathered data from these gauges for each portion of the bay and calculated the mean, median, 5th, and 95th percentiles for each
simulation category to visualize trends in surge timing and location. This data allows us to observe local changes, such as wind
setup and setdown where each water level gauge is located.

### 2.6.2 Inundation

We calculated inundation differences by first identifying the grid cells above MSL in the bathymetry (dry cells) that were
inundated (wet cells) in our baseline no-breach scenario. We used these wet cells to identify inundation changes for each
simulation. The changes in wet cells from our baseline scenario allowed us to see differences in inundation directly caused by
island breaching. Each cell covers 324 square meters (18x18 meters). We used GeoClaw's fgmax functionality, which tracks



**Table 2.** Peak surge arrival time in minutes from landfall for each synthetic water level gauge in Fig. 1 for each category of simulation

|         | Width | Depth | Width and Depth | Locations | East of Inlet | West of Inlet |
|---------|-------|-------|-----------------|-----------|---------------|---------------|
| Gauge a | 94    | 99    | 209             | 13        | 94            | 13            |
| Gauge b | 60    | 58    | 5               | 15        | 64            | 15            |
| Gauge c | 42    | 42    | 49              | 28        | 42            | 41            |
| Gauge d | 41    | 43    | 48              | 29        | 44            | 38            |
| Gauge e | 85    | 82    | 97              | 35        | 39            | 98            |
| Gauge f | 93    | 91    | 100             | 42        | 42            | 101           |

and updates the maximum surge in each cell until it reaches a peak. Unlike synthetic water level gauge data, these maximum surge and inundation values are snapshots in time and do not allow for dynamic analysis.

### 2.6.3 Bay Surge

We divided the bay into three sections (west, central, and east) and used our recorded maximum surge data to create surge distributions for each section by simulation category. We also analyzed the overall maximum surge for each bay section and generated maps that illustrate the flooding patterns across the entire region.

## 3 Results

We analyzed surge height and timing using synthetic water level gauges at specific bay locations. We divided the bay into three roughly equal longitudinal sections with two randomly chosen water level gauges per section shown as lettered circles in Fig. 1. Figure 4 presents these results showing the mean mean (solid lines), median (dashed lines) and 5th-95th percentiles (shaded areas) for each simulation category at each gauge location.

In all three bay sections, the peak surge arrives earliest in the *Location* scenarios, followed by the *West* scenarios in the west segment and the *East* scenarios in the central and east segments. The peak surge arrival time for each scenario and gauge is listed in Table 2. With the exception of the *Width and Depth* scenarios for Gauge b the *Location* surge arrives first. The *Location* surge is also larger for all gauges as seen in Fig. 4. In the west portion of the bay the *Location* and *West of Inlet* scenarios have similar maximum surge. Lastly, the maximum surge recorded by the eastern gauges is more consistent across all scenarios. In all cases the breaching simulations differ from the no-breach case.

Table 3 is a comparison of the L2 norms for the water level differences across gauge locations. To highlight the magnitude of variability in simulation category we calculated the differences between the 5th and 95th percentiles. The *Width and Depth* simulations showed the least variability in the west segment, while the *Location* simulations have the greatest variability across all gauge locations. The *East of Inlet* and *West of Inlet* water levels exhibit the least variability in the regions opposite the breaches.

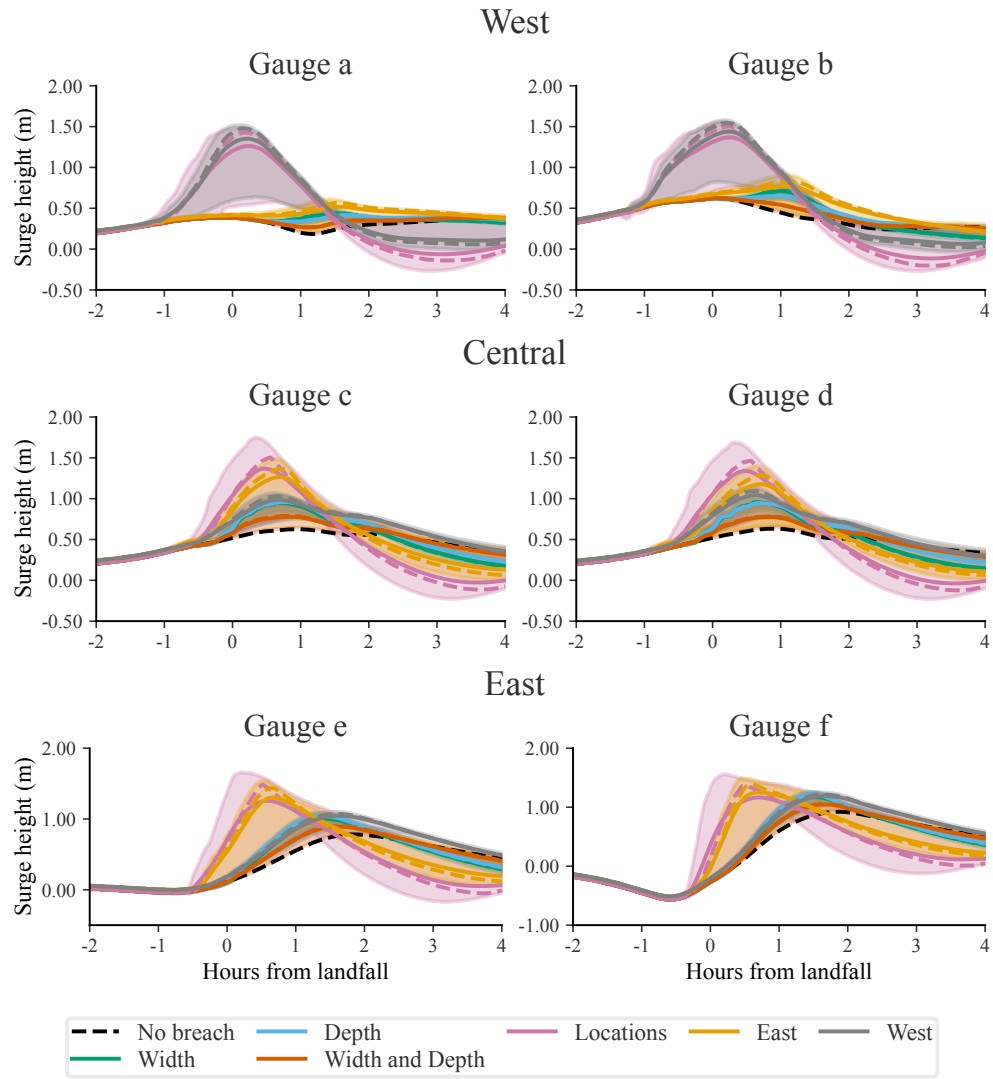

**Figure 4.** Synthetic water level gauges for each section of the bay. See Fig. 1 for locations. Each dark line is the mean of all of the simulations in that category, the dotted lines in each color represent the median of that category. Each shaded area covers the 5 - 95 percentile of the category

Figure 5 illustrates the maximum surge heights for each category of simulation at locations shown as green circles in Fig. 1. The *Location* simulations have the highest surge with density peaks at 1.85 meters, 1.75 meters, and 1.51 meters for west, central, and east bay respectively.

The standard deviation is also large at 0.32 m, 0.24 m, 0.26 m compared to the other simulations. When constrained to the original six breaches the *Depth* variations have the largest mean surge height at 1.06 m, 1.07 m, 0.84 m, with medians similar




**Table 3.** Comparison of L2 norms representing the differences in water level percentiles (5th and 95th) across different gauge locations.

|  | Width | Depth | Width and Depth | Locations | East of Inlet | West of Inlet |
|---|---|---|---|---|---|---|
| Gauge a | 10.07 | 6.19 | 15.98 | 112.30 | 28.82 | 91.96 |
| Gauge b | 15.89 | 10.95 | 23.45 | 97.89 | 42.16 | 79.64 |
| Gauge c | 22.03 | 17.49 | 34.17 | 115.71 | 77.69 | 33.36 |
| Gauge d | 21.47 | 16.27 | 32.85 | 109.37 | 74.61 | 37.92 |
| Gauge e | 23.54 | 20.38 | 34.19 | 138.75 | 96.81 | 29.06 |
| Gauge f | 22.77 | 19.73 | 33.06 | 156.11 | 103.66 | 28.14 |

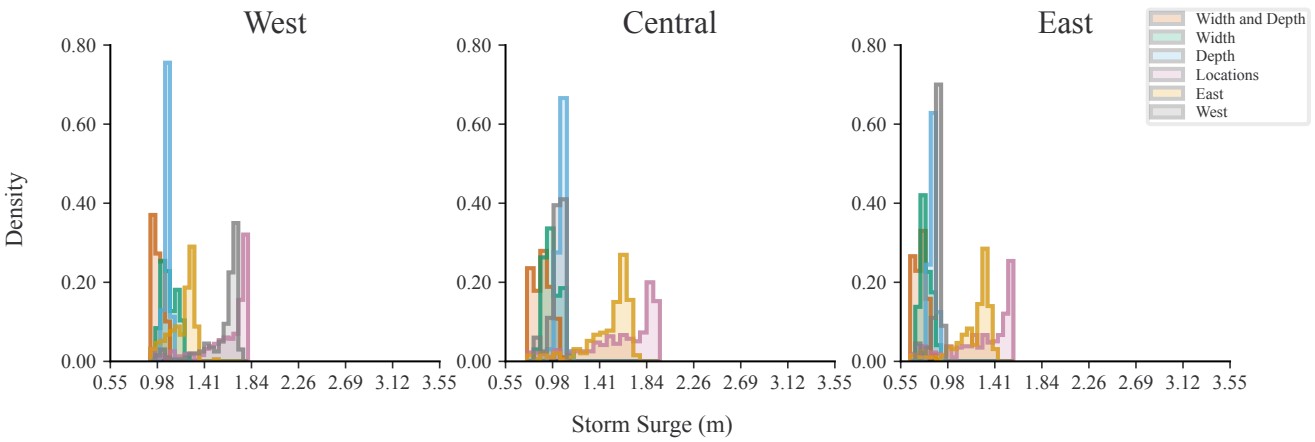

**Figure 5.** Maximum surge height in meters for each selected location shown in Figure 1 (green dots). Data shows 1900 scenarios split into six categories. Six breaches where width is randomized (blue) (464 scenarios), six breaches where depth is randomized (green) (424 scenarios). Varying width, depth, and number of breaches up to six breaches (orange) (297 scenarios). Varying width, depth, location, and number of breaches up to 295 breaches (315). Varying width, depth, number of breaches up to 100 but limiting breach location to the east of the inlet (yellow) (200) and west of the inlet (grey) (west

to the mean for all scenarios. The *Width* scenarios mean surge height is 0.97 m, 1.09 m, 0.78 m and the *Width and Depth* simulations have a mean surge height of 0.88 m, 0.98 m, 0.72 m. The *East of Inlet* simulations mean is 1.49 m, 1.20 m, 1.21 m, and the mean of the *West of Inlet* simulations is 1.02 m, 1.57 m, 0.88 m. Excluding the *Location*, *West*, and *East* simulations the largest standard deviation varies between the categories. The largest standard deviation is 0.08 m at the west location for the *Width and Depth*, the central standard deviation is 0.07 for the *Width* category, and east has a maximum standard deviation of 0.05 m which includes both *Width and Depth* and *Width*. Table 4 reports all of the results for the surge calculated by GeoClaw.

Figure 6 compares the maximum surge for each section of the bay. The overall surge distribution pattern is similar to those at the specific bay locations in Fig. 5. The *Location* simulations have the largest impact on surge for each bay section. The next



**Table 4.** Maximum surge height (m) for each category of breach simulations at each of the three points shown on Fig. 5

|  | Width | Depth | Width and Depth | Location | East of Inlet | West of Inlet |
|---|---|---|---|---|---|---|
| **west** |  |  |  |  |  |  |
| Min | 0.83 | 0.79 | 0.75 | 0.75 | 0.79 | 0.81 |
| Max | 1.13 | 1.12 | 1.07 | 1.96 | 1.73 | 1.08 |
| Mean | 0.97 | 1.06 | 0.88 | 1.60 | 1.49 | 1.02 |
| Median | 0.96 | 1.07 | 0.89 | 1.69 | 1.57 | 1.04 |
| Standard Deviation | 0.07 | 0.04 | 0.08 | 0.32 | 0.21 | 0.06 |
| Variance | 0.00 | 0.00 | 0.01 | 0.10 | 0.05 | 0.00 |
| Density Peak | 0.94 | 1.09 | 0.90 | 1.85 | 1.62 | 1.05 |
| **central** |  |  |  |  |  |  |
| Min | 0.95 | 0.95 | 0.92 | 0.92 | 0.94 | 0.97 |
| Max | 1.25 | 1.12 | 1.10 | 1.80 | 1.52 | 1.74 |
| Mean | 1.09 | 1.07 | 0.98 | 1.60 | 1.20 | 1.57 |
| Median | 1.08 | 1.07 | 0.98 | 1.70 | 1.24 | 1.66 |
| Standard Deviation | 0.07 | 0.02 | 0.05 | 0.24 | 0.11 | 0.18 |
| Variance | 0.00 | 0.00 | 0.00 | 0.06 | 0.01 | 0.03 |
| Density Peak | 1.04 | 1.07 | 0.93 | 1.75 | 1.28 | 1.67 |
| **east** |  |  |  |  |  |  |
| Min | 0.69 | 0.68 | 0.64 | 0.64 | 0.68 | 0.70 |
| Max | 0.90 | 0.88 | 0.84 | 1.58 | 1.41 | 0.93 |
| Mean | 0.78 | 0.84 | 0.72 | 1.30 | 1.21 | 0.88 |
| Median | 0.77 | 0.85 | 0.73 | 1.38 | 1.28 | 0.90 |
| Standard Deviation | 0.05 | 0.02 | 0.05 | 0.26 | 0.17 | 0.05 |
| Variance | 0.00 | 0.00 | 0.00 | 0.07 | 0.03 | 0.00 |
| Density Peak | 0.75 | 0.86 | 0.74 | 1.51 | 1.31 | 0.91 |

largest surge vary by bay section with *West of Inlet* having the largest surge in the west and the *Width* scenarios dominating the central and east regions.

Figure 7 illustrates the relationship between total breach area ($km^2$) and total inundation change from a no-breach simulation. All simulations except for *East of Inlet* and *West of Inlet* exhibit a logarithmic relationship that starts with a minimum inundation of 0.1632 km$^2$ and illustrates that a larger breach area leads to more inundation up to approximately 75 breaches where the 250 curve levels off at a total breach area of 0.035 km$^2$ and a maximum inundation area around 40.1 km$^2$. Beyond this point, inundation change increases more slowly to a maximum of 49.06 km$^2$. The second curve from the top represents the *West of*





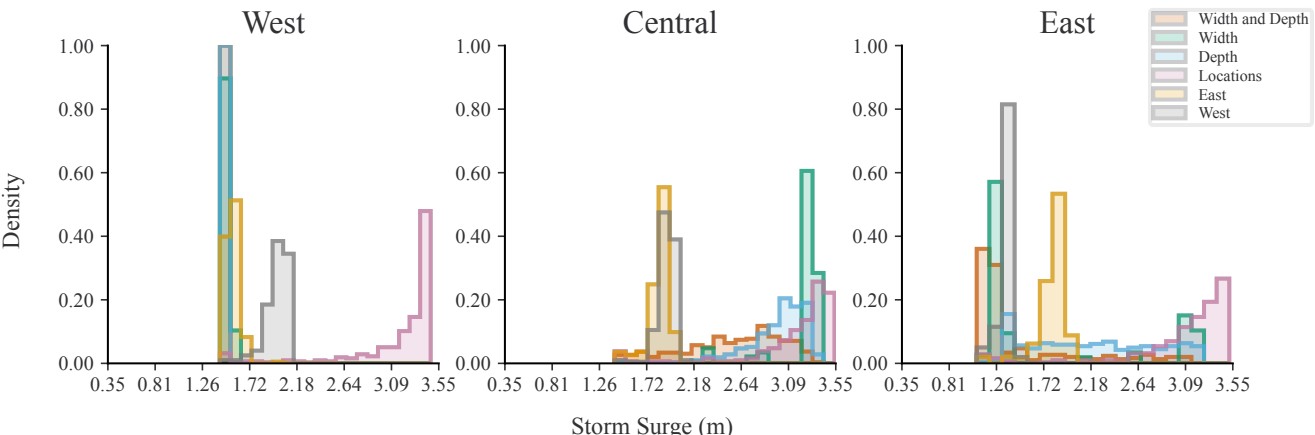

**Figure 6.** Maximum surge height (m) for each category of breach simulations across each entire section of the bay

*Inlet* scenarios. These simulations still show a logarithmic trend, but level off at a lower total inundation. The third curve is for the *East of Inlet* simulations, also leveling off at a lower total inundation.

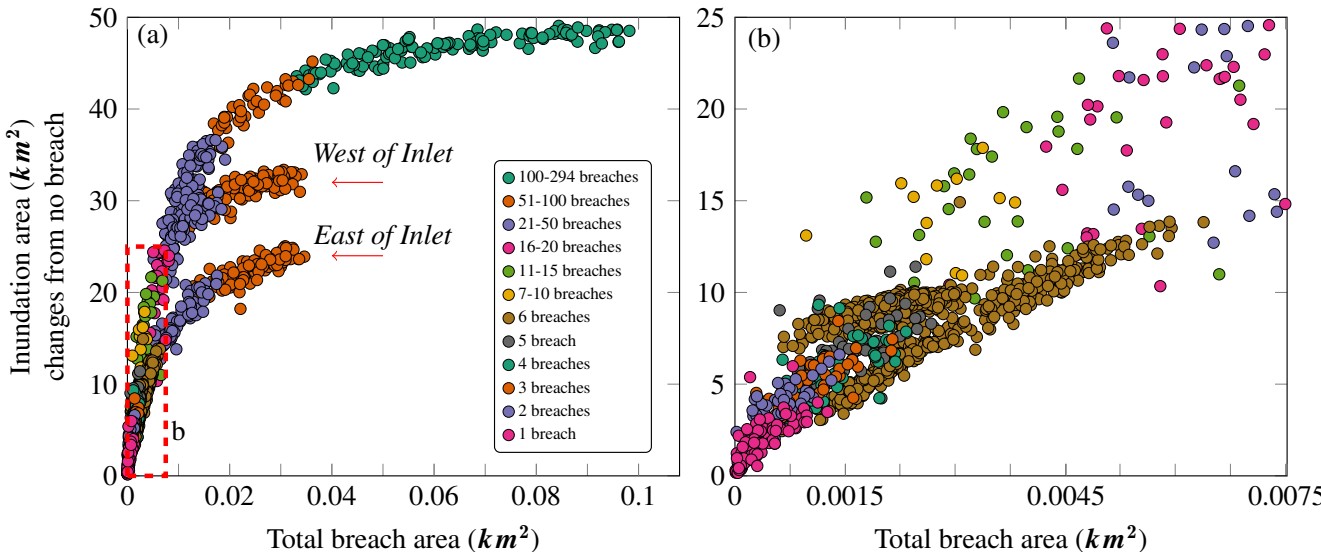

**Figure 7.** a) Total inundation vs. total breach size for all 1900 scenarios, points are colored per number of breaches. b) zoom in of a) panel to show differentiation of breach area and number of breaches and how the inundation can vary





## 4 Discussion

The results of this study show that the location, size, and number of breaches affect coastal flooding. There is a clear relationship between total breach area and flooding in the bay and on the mainland coastline. The histograms in Fig. 5 illustrate various bay locations. The west surge point near the Forge River mouth shows that, with the original breach locations, the surge distribution is clustered and overlaps across scenarios. However, when breach locations vary, the maximum surge is much higher from nearby breaches. This location initially experiences surge from the eastern side of the bay, and after landfall, surge

is pushed again from the western connection to Great South Bay and the nearby breaches. Figures 8 and 9 further illustrate that the maximum surge in the bay and along the mainland alters due to breach location. We further discuss the surge patterns of Fig. 8 and Fig. 9 below.

The central location is adjacent to the mainland coastline near Seatuck Cove. Varying breach locations and numbers increases surge, but to a smaller degree than in the west. The maximum surge here is lower than in the west likely due to its proximity to

265 the inlet. The peak surge western breaches reaches the inlet before Seatuck Cove, allowing water to exit the bay and reducing the surge. Additionally, the coastline's shape helps protect this location from surge coming from the southwest. However, *East of Inlet* breaching is the second largest contributor of surge at this location. Surge from eastern breaches is directed westward by hurricane wind circulation impacting the central location. Figure 10a displays surge patterns from eastern island breaching.

The east location, which is closest to the barrier island, has the smallest maximum surge in the bay. Breaches formed eastward

of the original locations do bring more surge to this location. However, its proximity to the inlet means that bay surge traveling east after peak ocean surge flows out of the inlet, reducing the total surge. Additionally, wind pushes the eastern surge towards the southwest, further reducing the total surge.

Figure 4 shows how surge timing and maximum surge vary across scenarios for each bay section. Many *Location* simulations have breaches in the southwest portion of the barrier island. The peak ocean surge spreads from the southwest to northeast

before landfall causing these breaches to open earlier than in other scenarios. This is evident in west gauges (a and b), where the surge arrives earlier and is larger than simulations with breaches closer to the inlet. The central gauges illustrate that while the *Location* surge remains larger, its timing is more aligned with other scenarios. The eastern gauges maximum surge is not much higher than the other categories, but the surge still arrives earlier due to water entering the bay from the southwest breaches.

Additionally, Fig. 4 shows a change in surge direction in the no-breach scenario. Around landfall, the surge direction is from northeast to southwest, and reversed shortly after where water is pushed from the connection to Great South Bay. This created a local setdown for gauges not protected by the mainland's contours. Gauges a, e, and d show this setdown, seen in gauge a, approximately one hour after landfall and in in gauges c and d 30 minutes later. The *Width and Depth* simulations also exhibit this setdown, having the least impact on bay surge. Two to three hours post-landfall, the breaches allow water to flow back

into the ocean, leading to a more rapid reduction in bay flooding than in the no-breach scenario. We see this especially in the *Location*, *East of Inlet*, and *West of Inlet* simulations where the water recorded at the gauges are reduced below the no-breach scenario.



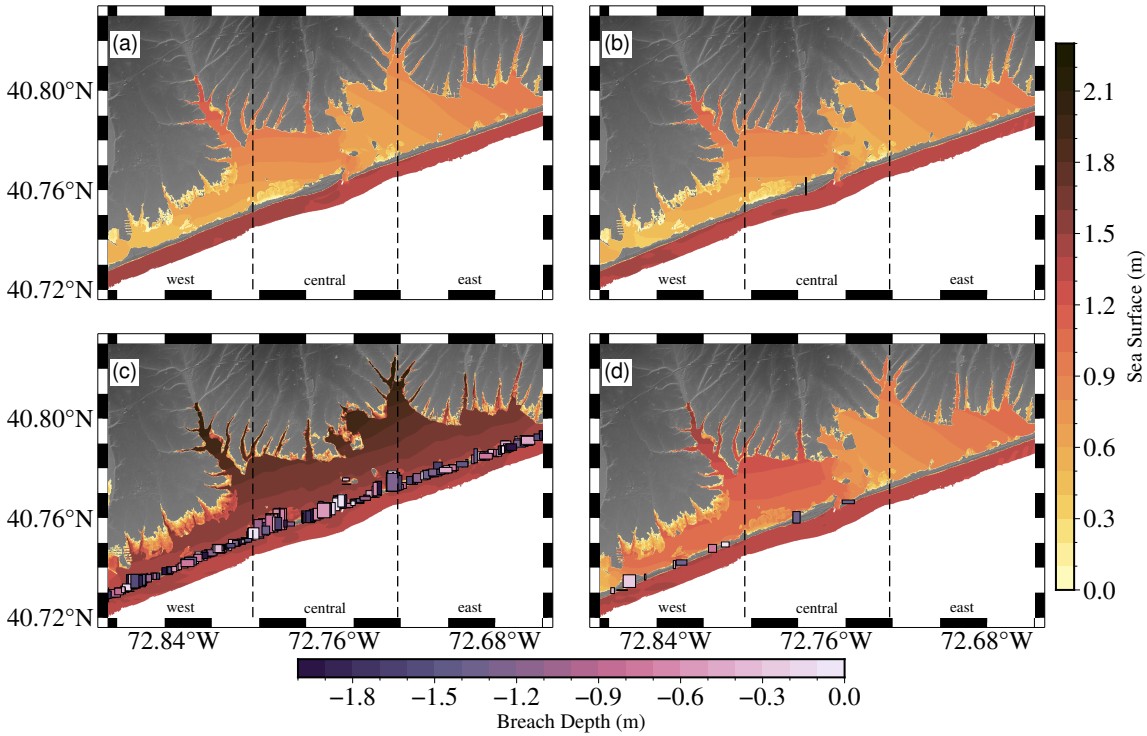

**Figure 8.** Maps Moriches Bay, NY. Each panel is a separate simulation representing different values of storm surge inundation. Panel (a) is our no-breach scenario. Panel (b) is the minimum inundation of 0.162 $km^2$ with a single small breach. Panel (c) is the largest inundation scenario of 49.06 $km^2$ with 259 breaches. Panel (d) is a simulation that has the closest inundation to the mean of all 1900 simulations 14.54 $km^2$, with eleven breaches.

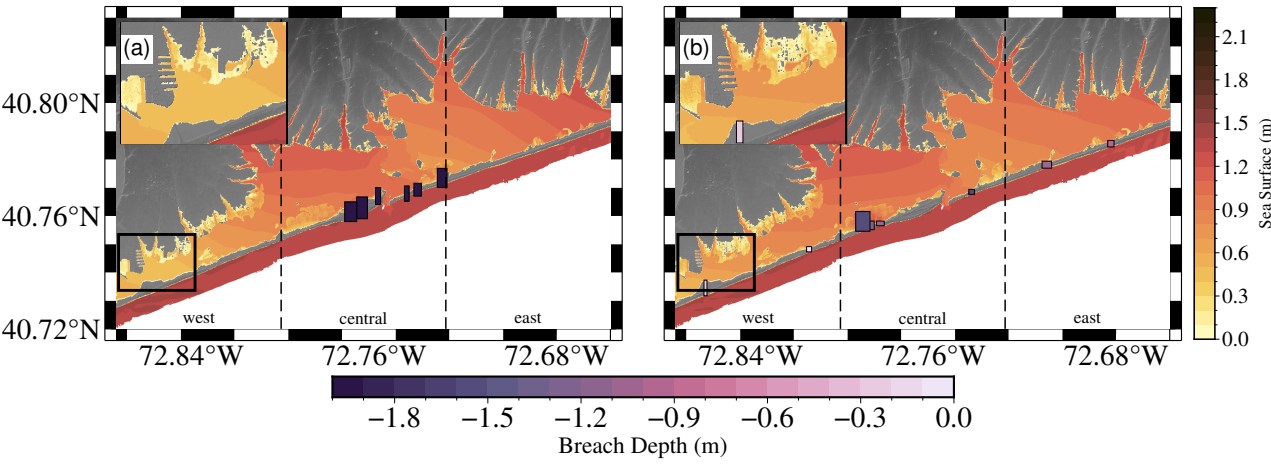

**Figure 9.** a) Maximum surge and inundation for simulations with 6 breaches and total breach area 0.0039 $km^2$. b) Maximum surge and inundation for simulation that has 11 breaches and a total breach area of 0.0036 $km^2$.



In Fig. 7a we show that the total breach dimensions are related to the total area of inundation, with larger and more numerous breaches bringing more water inland. Total breach area across all breaches is the strongest predictor of coastal inundation, until the island is significantly eroded, after which inundation growth slows considerably. Figure 7b adds nuance to this relationship. While there is a stronger correlation between breach width and inundation than depth and inundation, the maximum breach depth of two meters is at least a factor of 20 smaller than total width for these scenarios, reducing the impact of depth on the hydrodynamics of each breach. The cluster of breaches above the main group are the *Depth* scenarios, whereas the *Width* scenarios exhibit a more linear relationship with total inundation area. The secondary and tertiary curves in Fig. 7a show that inundation is capped if half of the barrier island still exists. These simulations were isolated to only the west and east sides of the inlet. Both scenarios have a smaller total impact on inundation; however, the *West of Inlet* scenarios contribute to a larger total inundation than the *East of Inlet* breaches. This is because breaching starts earlier in the west due to the storm's approach direction, and water moving southwest out of the inlet relieves some of the inundation that accumulates on the east.

Figure 8 illustrates that different inundation and bay surge patterns correlate with the number and size of breaches. Panel a) depicts a no-breach scenario, which is similar in surge and inundation distribution to the minimum inundation (panel b) featuring a single small breach. The difference between these simulations is approximately 500 wet vs. dry cells, which is 163,200 m$^2$ (0.1632 km$^2$) of inundation. Panel d) represents a simulation with eleven medium sized breaches, totalling 0.0029 $km^2$ in breach area and 14.54 $km^2$ in inundation change, closely approximating the mean inundation change from all 1900 simulations. The simulation closest to the median of the scenarios (not shown) has six medium sized breaches and 9.36 $km^2$ of inundation. The surge contours in the bay and the total horizontal coastal inundation are very different from the no-breach or minimum inundation scenarios, with higher flooding potential in the coves, creeks and rivers that border the bay and along the lower elevation coastlines. The maximum inundation scenario (panel c) where most of the island has been breached, shows a bay surge of approximately two meters resulting in complete flooding of the lowest elevation areas of the coastline.

The impact of differing breach locations on inundation as illustrated in Fig. 8c, can be further seen in Fig. 9, which compares two simulations with a similar total breach area, but different total inundation. Figure 9a, shows a scenario with six moderately sized breaches in the locations from the 1938 hurricane, with a total breach area of 0.0039 $km^2$, and total inundation of 10.44 $km^2$. In contrast, Figure 9b, has a smaller total breach area of 0.0036 $km^2$ but a larger inundation at 12.03 $km^2$. In this scenario the breaches are generally smaller but more spread out across the barrier island, with closer to Great South Bay in the western portion. While the bay surge patterns are similar, the surge contours differ, and the breaches between Great South Bay and Moriches Bay allow more water to flow in from the southwest prior to peak ocean surge. This results in more coastal inundation along the western coastline (see inset of Fig. 9a and b). The Forge River surge is higher in Fig. 9b and the inlet region has a lower total water depth. The eastern portion of the bay has less inundation in Fig. 9b, likely due to most breaches being in the west, the inlet allowing water to flow out, and the lower elevation of the western half of the bay's coastline.

Breaching isolated to one side of the inlet creates notable changes in bay surge and total inundation, as shown in Fig. 10. As Fig. 7 indicates, the *West of Inlet* scenarios result in higher total coastal inundation which is evident in Fig. 10b where the western mainland coastline is significantly inundated compared to other scenarios. In contrast, the *East of Inlet* simulations can





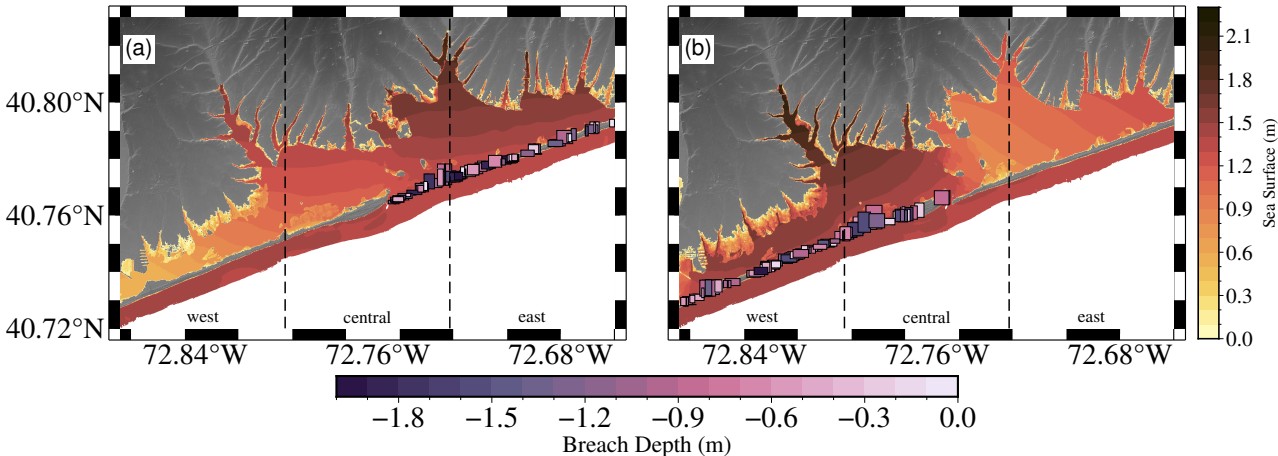

**Figure 10.** a) Maximum surge and inundation for *East of Inlet* simulations with a total inundation of 24.87 $km^2$. b) Maximum surge and inundation for *West of Inlet* simulation that has a total breach area of 26.85 $km^2$.

push the surge further down the bay. As the storm continues past landfall the surge is pushed southwest and not all of it floods out through the inlet.

Figures 8, 9, and 10 highlight the key findings from our simulations. Total breach area is a strong predictor of total inundation;
however, breach location is also crucial, especially given the storm's forcing dynamics and surge direction. While this study does not include tides and waves, they significantly influence bay surge dynamics and contribute strongly to breach initiation and growth. The stochastic nature to these processes makes them difficult to model, and much of our understanding relies on empirical observations. Incorporating tidal or wave components into our simulations could result in different patterns of breaching and inundation. Our use of offshore water levels to model breaching assumes wave action contributes to breach
initiation, based on prior studies and observations.

## 5 Conclusions

Breaching of a barrier island during a hurricane shows a strong impact on mainland inundation. The number, locations, and size of the breaches can change the inundation potential for the coastline. Understanding vulnerable areas and how breaching impacts them can provide opportunities for shoring up infrastructure and allowing planning that minimizes the storm's disruption
to lives and the community.

Ideas for future work that can expand this study include increasing the number of simulations to better refine the statistical distribution of the different breaching simulations. It will take significantly more data to find a consensus on specifically vulnerable locations, patterns of breaching, and its coastal impacts. Repeating simulations but varying the storms will also provide insight into how the approach, speed, landfall location, and size of the storm affects the flooding and inundation.
Performing additional simulations for other barrier islands, storms, and coastlines not only helps local communities plan for





their own vulnerabilities but performing large scale simulations is valuable for expanding understanding of commonalities and differences of different regions.



**Code availability section**

BarrierIslandBreachProject Developer: Catherine R Jeffries Contact: catherinej@vt.edu, 253-670-8643

Program language: Python, FORTRAN

Software required: GeoClaw, Linux or MacOS

The source codes are available for downloading at the link:

Code for preparation and analysis (available 2024) https://github.com/cjeffr/BarrierIslandBreachProject

GeoClaw v5.9.2 available(2023) (https://www.clawpack.org/installing.html)

Modifications to GeoClaw to run breaching and storm inputs:

https://github.com/cjeffr/geoclaw/tree/OWI_integration

*Author contributions.* Robert Weiss, Jennifer L. Irish, and Kyle Mandli conceptualized this research, provided supervision, and reviewed and edited the manuscript. Catherine R. Jeffries developed the methodology and software needed for this project, performed the validation, formal analysis, and figure creation, and wrote the original draft of the manuscript.

*Competing interests.* The authors declare that they have no conflicts of interest.

*Acknowledgements.* The authors would like to acknowledge the US Army Corps of Engineers (USACE) for providing the storm forcing data used in this study.

The authors acknowledge Advanced Research Computing at Virginia Tech for providing computational resources and technical support that have contributed to the results reported within this paper. URL: https://arc.vt.edu/

**Funding Acknowledgements** This material is based upon work supported by the National Science Foundation under Grant Number 1735139. Any opinions, findings, and conclusions or recommendations expressed in this material are those of the authors and do not necessarily reflect the views of the National Science Foundation. This work was supported in part by the National Science Foundation [Grant Number 1735139] and the Institute for Critical Technology and Applied Sciences (ICTAS), Virginia Tech.

During the preparation of this work the author(s) used ChatGPT in order to reduce the word count of the document to fit into the journal's
length restrictions. After using this tool, the author(s) reviewed and edited the content as needed and take(s) full responsibility for the content of the publication.



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
