# Peer review of "Impacts of Barrier-Island Breaching on Mainland Flooding During Storm Events applied to Moriches, NY"

_EGUsphere, 2024_

## Author Comment (AC1)

January 5, 2025.

**1    Author remarks**

Thank you for taking the time to review this paper and offer suggestions for improving the quality of the research and writing. We have worked our way through the reviewer comments and addressed them below. Under each point brought up by the reviewers we have made comments addressing their concern (grey boxes) and added or updated the paper. Our added or updated text is in blue with strikeout for removed text and underlines for added text. In Green are references to the line numbers, both original and revised manuscripts. If only Revised is listed that is an indication that this is added text.

> Author reply to comments

>  and added text

> Line numbers from Original and Revised manuscripts

**Reviewer 1**

**1.1.** The Discussion is a very well written interpretation of the results, but lacks any connectivity to other research. It is unbelievable that not a single reference is cited in the Discussion! I urge the authors to compare their results and interpretations with the work of others who have conducted similar analyses in coastal barrier island systems. The authors could also compare their findings with the findings from other sandy coastal systems, and from other types of coastal hazards.

> Thank you for your comment. We have updated our discussion to include references to other works that our study can be compared with

[revised manuscript text omitted]

 Figs. 8, 9, and 10 highlight the key findings from our simulations. Total breach area is a strong predictor of total inundation; however, breach location is also crucial, especially given the storm's forcing dynamics and surge direction. Similarly, a study by Gharagozlou et al. on breaching's impact on lagoon circulation during Hurricane Isabel illustrates how breaches alter flow patterns and introduce larger volumes of ocean water into the lagoon. These findings can be compared to our results, which demonstrate that breach location significantly influences storm surge behavior and its subsequent effects on coastal flooding Gharagozlou et al. (2021). While this study does not include tides and waves, they significantly influence bay surge dynamics and contribute strongly to breach initiation and growth as described in Smallegan and Irish (2017); Sherwood et al. (2014); Safak et al. (2016). The stochastic nature to these processes makes them difficult to model, and much of our understanding relies on empirical observations from geological studies or post-storm surveys of barrier island systems Kraus et al. (2002); Buynevich and Donnelly (2006). Incorporating tidal or wave components into our simulations could result in different patterns of breaching and inundation. Our use of offshore water levels to model breaching assumes wave action contributes to breach initiation, based on prior studies and observations.

Original: 255 - 330
Revised: 263 - 346

**1.2.** line 12 we've

The insights  we have gleaned from this study can help prepare shoreline communities for the differing ways that breaching affects the mainland coastline.

> Original: 11 - 13
> Revised: 11 - 13

**1.3.** 50 space between ), these

> Thank you for pointing this out, we have added a space

**1.4.** 51 formatting incorrect, check latex

> Thank you for showing me this error I have changed it to an in text citation

> Lab and field experiments by  Visser (1999) for breaches in dikes are useful but the breach is initiated with a pre-drilled hole in the dike and does not simulate exactly what occurs to barrier islands during storms.

> Original: 51
> Revised: 52

**1.5.** 80 what is the difference between Moriches and Moriches bay?, could these be labeled in fig 1?

> Moriches is a small region in Suffolk county New York, Moriches Bay is the bay in the same region that separates the mainland coastline from the barrier island that parallels the coastline.

> We updated the map to include new annotations for other regions, Moriches, NY (the small town) is under the inset

**1.6.** 81 please provide detail on the damage

> We have added details on the damage from the 1938 hurricane

> The 1938 Hurricane made landfall as a category 3 hurricane near Moriches, NY on September 21, 1938. The maximum sustained wind speed recorded during this hurricane was 178 km/hr at Blue Hill Observatory, MA (Brooks, 1939). The center of the storm passed over the western side of Great South Bay, less than 75 km from Moriches Bay. Figure 1 illustrates the location of Moriches Bay, NY and the storm's location as it made landfall. It generated 10 breaches across the barrier island system and caused widespread damage (Morang, 1999; Coch, 1994; Cañizares and Irish, 2008). The damage caused by this hurricane included 564 deaths, widespread flooding from both storm surge and high rainfall amounts, thousands of structures were damaged or destroyed and

> widespread power outages across southern New England (Vallee and Dion, 1997) Six breaches were opened during the hurricane at Moriches, NY specifically, three each on either side of the inlet (Howard, 1939). Aerial photographic evidence illustrates that widespread breaching of the island occurred during the storm (Howard, 1939) and described the breaches as widening of the original inlet which was opened in 1931. These breaches were closed after the storm but the timeline and method for closure is unclear in the literature (Cañizares and Irish, 2008; Howard, 1939)

> Original: 88 - 92
> Revised: 90 - 100

**1.7.** Figure 1 Is this a lidar image? What is the vertical scale? The storm track on the inset can be more pronounced, please label the features, Moriches bay, various islands in the main image, what is the weird cross hatching in the barrier island between b and c

> This is an image created from a topobathy dataset that was partially created with lidar yes. The cross hatching is a part of a manmade series of channels to control mosquito populations, (). Please see the updated figures below for figure edits

**1.8.** 91-92 what is the evidence for this? How quickly did they close after the hurricane? Please explain the significance these breaches were for flooding. Please show the breaches location on figure 1

> Please see updated figure 1, We included the original breach locations as stars on the figure. We've added a citation to discuss the breaches that formed, See comment 1.6 above for the revised sentences and citations

**1.9.** 100 in what case? Recommend deleting

> thank you for pointing out this interesting phrasing, I have corrected the sentence

> GeoClaw has been validated by the US National Tsunami Hazard Mitigation Program (NTHMP) for tsunami modeling. González et al. (2011) describes the benchmarking process used to validate GeoClaw .

> Original: 99 - 100
> Revised: 106 - 108

**1.10.** 102 spell out adcirc then use acronyms

> thank you for pointing this out, it is fixed

Storm surge modeling with GeoClaw has been proposed to provide a robust but less computationally expensive model than  the ADvanced CIRCulation model (ADCIRC), a commonly utilized finite element model (Westerink et al., 2008; Mandli and Dawson, 2014; Bates et al., 2021).

Original: 101-103
Revised: 109 -111

**1.11.** figure 5 hanging sentence, west - ?

This was a typo, it has been fixed

**Reviewer 2**

**2.1.** unify fonts and scales, size of text in figs 8,9,10 also seems odd the lat/lons are in larger font than scale and features annotate to highlight key features

Thank you for pointing on the inconsistencies in the figures. They have been updated to share a common scale and font and additional features added to Figure 1

See below for all updated figures

**2.2.** 30-33 awkward sentence

Thank you for pointing out the awkward section, we have updated the wording.

Storm surge that  creates a water level gradient between the ocean and back-barrier region forces water to flow rapidly over the barrier island  . This flow erodes sediment in an effort to equalize the water  gradient involves a critical elevation of water  that can cause significant erosion without inundating the island (Kraus et al., 2002; Kraus, 2003).

Original: 30 - 35
Revised: 30 - 35

**2.3.** 40, 44 reference format

Thank you for pointing out the reference issues. We have fixed these to be in text citations

**2.4.** 46 at and at in same sentence

Thank you for showing this grammar issue. We have fixed it per below

The sediment transport model generated two breaches at  peak erosion sites, but neither breach location matched the observed breach that opened during Hurricane Sandy (van der Lugt et al., 2019).

Original: 46 - 48
Revised: 47 - 49

**2.5.** 53 just geological mapping, no need for 'a'

Buynevich and Donnelly (2006) performed  geologic mapping of some New England, USA barrier islands and found geologic signatures to indicate the islands' past history with breaching and overwash.

Original: 53 - 54
Revised: 55 - 57

**2.6.** 83 change especially to a real word like particularly? C2.6

Thank you for suggesting this change. We have updated it.

Our study focuses on Moriches, NY a section of the barrier island that spans Long Island, New York, USA along the Atlantic Ocean. This region is heavily populated and is regularly impacted by storms.  , in particular, the 1938 hurricane caused extensive damage at Moriches, NY.

Original: 83
Revised: 81 - 84

**2.7.** 88 more details on max winds

We've added the maximum wind speeds and a citation

The 1938 Hurricane made landfall as a category 3 hurricane near Moriches, NY on September 21, 1938. The maximum sustained wind speed recorded during this hurricane was 178 km/hr at Blue Hill Observatory, MA (Brooks, 1939) . The center of the storm passed over the western side of Great South Bay, less than 75 km from Moriches Bay.

> Original: 88 - 89
> Revised: 90 - 92

**2.8.** 159 should be computation time

> We have updated this typo

> We restricted GeoClaw's adaptive refinement for Moriches Bay and its adjacent barrier island to an 18 x 18 meter grid, balancing resolution with  computation time.

> Original: 159 - 160
> Revised: 166 - 168

**2.9.** 282 awkward sentence

> This setdown is observed on Gauges a, e, and d  at approximately one hour after landfall  ulineon gauge a, and 30 minutes later on gauges e and d.

> Original: 282 - 283
> Revised: 291 - 293

[Figure]

Figure 1: Map of study area Moriches Bay, NY. Inset shows region surrounding Moriches Bay, NY (orange box). Storm track for the 1938 Hurricane (light blue line) and our simulated storm. Green circles are locations of surge measurements from Table 2. Remaining circles are locations of synthetic water level gauges illustrated in Fig. 4. Original breach locations marked by stars

**3 Figures**

[revised manuscript text omitted]

Safak, I., Warner, J. C., and List, J. H. (2016). Barrier island breach evolution: Alongshore transport and bay-ocean pressure gradient interactions. *Journal of Geophysical Research: Oceans*, 121(12):8720–8730.

Sherwood, C. R., Long, J. W., Dickhudt, P. J., Dalyander, P. S., Thompson, D. M., and Plant, N. G. (2014). Inundation of a barrier island (chandeleur islands, louisiana, usa) during a hurricane: Observed water-level gradients and modeled seaward sand transport. *Journal of Geophysical Research: Earth Surface*, 119(7):1498–1515.

Smallegan, S. M. and Irish, J. L. (2017). Barrier island morphological change by bay-side storm surge. *Journal of Waterway, Port, Coastal, and Ocean Engineering*, 143(5):04017025.

Vallee, D. R. and Dion, M. R. (1997). Southern new england tropical storms and hurricanes: a ninety-seven year summary, 1900-1996, including several early american hurricanes.

van der Lugt, M. A., Quataert, E., van Dongeren, A., van Ormondt, M., and Sherwood, C. R. (2019). Morphodynamic modeling of the response of two barrier islands to atlantic hurricane forcing. *Estuarine, Coastal and Shelf Science*, 229:106404.

Visser, P. J. (1999). Breach erosion in sand-dikes. In *Coastal Engineering 1998*, pages 3516–3528.

Visser, P. J. (2001). A model for breach erosion in sand-dikes. In *Coastal Engineering 2000*, pages 3829–3842.

Westerink, J. J., Luettich, R. A., Feyen, J. C., Atkinson, J. H., Dawson, C., Roberts, H. J., Powell, M. D., Dunion, J. P., Kubatko, E. J., and Pourtaheri, H. (2008). A basin-to channel-scale unstructured grid hurricane storm surge model applied to southern louisiana. *Monthly weather review*, 136(3):833–864.

---

## Author Response (AR2)

June 5, 2025.

**1 Author remarks**

Author reply to comments

 and added text

Line numbers from Original and Revised manuscripts

**Reviewer 1**

**1.1.** The authors have done a reasonable job in addressing the initial comments, particularly in clarifying aspects of methodology and improving the structure of the manuscript. The case study presented is well-articulated and the data are clearly analysed and interpreted.

However, concerns remain regarding the lack of international context. At present, the manuscript is limited to a U.S.-based case study, with little reference to comparable studies elsewhere. To strengthen the manuscript's broader relevance and scholarly contribution, the authors are encouraged to add a paragraph discussing barrier breaching and overwash studies from other regions. Some suggested references include Sánchez-Arcilla and Jiménez (1994) and Zăinescu et al. (2019) for European examples (there are many), Noval (2024) for Brazil, Soria et al. (2020) for the Philippines, and Gouramanis et al. (2025) for India. These studies provide valuable comparative insights and may include additional references worth integrating.

Incorporating this international perspective would improve the manuscript's depth and relevance. It is recommended that the authors return to the global literature in the second or third paragraph of the discussion, and again in the conclusion, to reflect on how their findings align with or diverge from those in other geographic contexts. I think this will greatly enhance the paper's appeal to an international audience and demonstrate its contribution to broader coastal geomorphological research.

Thank you for providing more insight into making this paper broadly applicable rather than isolated to a single case study. We have updated the text to include some of the suggested references and added additional references that matched our updated paper. The additions have been added to the introduction, discussion, and conclusion and the additions and their line numbers are below.

Barrier islands are long, shore-parallel, low-relief land masses that are adjacent to approximately 6.5% of the world's coastlines (Oertel, 1985; Stutz and Pilkey, 2001). According to Oertel (1985) barrier island systems consist of six sedimentary

environments; proximity to the mainland, a back-barrier region (bay or lagoon), an inlet and inlet delta, the barrier island, the barrier platform, and the shoreface. Barrier islands are protective structures that help dissipate wave energy and storm surge approaching the mainland from the ocean. Barrier island dunes that are higher than the approaching storm surge cause wave breaking, which reduces the impact of the surge when it reaches the back-barrier bay (Oertel, 1985; Irish et al., 2010). Vegetated low-lying dunes are more resistant to erosion and will absorb some of the seaward driven surge and wave energy. The dissipation of wave energy ensures that barrier islands undergo significant changes during storms and hurricanes, one of which is breaching. A breach is an opening in a narrow landmass, such as a barrier island, that allows a direct hydrodynamic connection between the ocean and the back-barrier bay or lagoon (Kraus, 2003; Wamsley and Kraus, 2005). Naturally occurring breaching is a complex process that involves the interaction of storm surge, waves and their resulting overwash with barrier island width, height, sediment characteristics, vegetation, and underlying geological structures. Storm forcing combined with local bathymetry is necessary to initiate the conditions that lead to breaching. Variations in storm size, intensity and locale may cause breaches in some locations but not in others.

During storm-induced overwash and inundation, water flowing across the island can scour a channel between the sea and the back-barrier region (Kraus, 2003; Pierce, 1970; Roelvink et al., 2009). This scouring requires strong flow and sustained inundation. Breaching can occur from both the seaward and landward side of the barrier island, but field data are limited in showing the direction of breach initiation (Kraus, 2003; Pierce, 1970; Smallegan and Irish, 2017). However, Smallegan and Irish (2017) show that bay surge following peak ocean surge is more likely to cause breaching from the landward side, as peak ocean surge already weakened the dune through erosion caused by wave attack and swash (Kraus, 2003; Smallegan and Irish, 2017). Identifying breach locations is challenging; localized lows for dune height and narrower portions of the island are more likely potential breach locations (Kraus, 2003; Kraus and Wamsley, 2003). One study by van der Lugt et al. (2019) simulated Hurricane Sandy (2012) using surge-tide levels, 2D wave-spectra, and sediment transport to model barrier island morphodynamics with pre-storm LiDAR bathymetric grids. The sediment transport model generated two breaches at peak erosion sites, but neither breach location matched the observed breach that opened during Hurricane Sandy (van der Lugt et al., 2019).

Studies focusing on coastal barrier morphodynamics during storms provide valuable insights into the erosion processes that can lead to breaching. Novak et al. (2024) demonstrate how the angle at which the storm surge and waves approach the island can exploit existing barrier vulnerabilities, resulting in extensive overwash deposits at the Paraíba do Sul River Delta Complex along the northern coast of Rio de Janeiro, Brazil. Their work emphasizes that vulnerability assessment must consider both event-scale storm characteristics and longer-term barrier

evolution patterns, as areas weakened by previous storms or longer-term barrier morphological changes can create points along the barrier that are more vulnerable to breaching and overwash. Similarly, analysis of a large breach along the Trabucador Bar in Spain led researchers to develop an erosion susceptibility index for different portions of coastal barriers, which incorporates barrier height, width, and offshore bathymetry to identify locations vulnerable to overwash and breaching (Sánchez-Arcilla and Jiménez, 1994). However, such indices require calibration for specific barrier geometries and wave climates, limiting their transferability across different coastal systems. The distinction between overwash and breaching represents a critical threshold in barrier response to storms. Analysis of overwash deposits serves as a critical tool for understanding storm-sediment dynamics in regions where direct storm observations are limited (Soria et al., 2021; Novak et al., 2024; Zăinescu et al., 2019; Donnelly et al., 2006; Houser et al., 2008; Matias et al., 2008). While these morphodynamic studies improve our understanding of where breaching may occur, we are still limited in data of breaching during actual storm events.

Quantifying breach dimensions during hurricanes is challenging. While breach growth over time has been documented (Kraus and Wamsley, 2003; Schmeltz et al., 1982), these studies focus on days to months post-storm. Predicting breach locations and tracking their growth during a hurricane is not feasible. Lab and field experiments by Visser (1999) for breaches in dikes are useful but the breach is initiated with a pre-drilled hole in the dike and does not simulate exactly what occurs to barrier islands during storms. Buynevich and Donnelly (2006) performed geologic mapping of some New England, USA barrier islands and found geologic signatures to indicate the islands' past history with breaching and overwash. Buynevich and Donnelly (2006) found ephemeral breaches with widths of 10 - 30 m before closing and breach depths of one - three meters below the dune crest. Some post-storm surveys have defined breach sizes before natural or forced closing. Kraus and Wamsley (2003) discusses Pike's Inlet on Long Island, New York, USA which was initially measured after the hurricane at 304.8 m wide and a nearby breach named Little Pike's Inlet was initially 30.48 m wide but over several months grew to over 914.4 m before it was closed. A breach near Moriches Inlet on Long Island studied by Schmeltz et al. (1982) had an initial size of 91.4 m and 0.61 m depth. This breach expanded to 885 m with a depth of three meters before it was mechanically closed. The uncertainties in breach dimensions and in where, how, and when breaches occur remains one of the many issues facing coastal communities today, due to the inability to predict or plan for the probable impacts of a breach forming where populations are highest.

Barrier islands are found along the coasts of 18 US states bordering the Atlantic Ocean and Gulf of Mexico (Zhang and Leatherman, 2011). As coastal populations have increased significantly in recent decades, the protective nature of barrier islands has become more crucial (Zhang and Leatherman, 2011). According to the

US National Hurricane Center (NHC), storm surge is the leading cause of loss of life and property damage during hurricanes (National Hurricane Center, 2006). Storm surge can cause flooding that damages structures, closes roads, and disrupts coastal communities. It can also accelerate erosion on barrier islands and the mainland, increasing flood risk. Understanding how barrier island breaching affects coastal flooding from storm surge is vital for risk assessment and mitigation. A hydrodynamic connection between the ocean and back-barrier region can lead to increased flooding and wave action during hurricanes, heightening risks to populations and property. However, there is little information on how different breach morphodynamics affect the mainland.

In this paper, we explore the different inundation patterns and surge depths at Moriches, New York, USA for a storm that approximates the 1938 Hurricane. Using GeoClaw, software capable of modeling storm surge, we artificially alter the bathymetry of a barrier island during a storm simulation to create breaches in the barrier island (Mandli et al., 2016). This method removes the complexities of modeling the morphological processes driving breach formation so we can purely study the coastal response to these breaches. We randomized the number, width, and depth of these synthetic breaches to gain a statistical understanding of how these parameters influence coastal inundation and bay storm surge.

Original: 15 - 80
Revised: 15 - 95

Fig. 4 shows how surge timing and maximum surge vary across scenarios for each bay section. Many *Location* simulations have breaches in the southwest portion of the barrier island. The peak ocean surge spreads from the southwest to northeast before landfall, causing these breaches to open earlier than in other scenarios. This  pattern is most evident in west gauges (a and b), where the surge arrives earlier and is larger than simulations with breaches closer to the inlet. The central gauges illustrate that while the *Location* surge remains larger, its timing is more aligned with other scenarios. The eastern gauges maximum surge is not much higher than the other categories, but the surge still arrives earlier due to water entering the bay from the southwest breaches. The strong correlation between breach locations and surge direction aligns well with findings from other coastal barrier studies. Novak et al. (2024), explain that surge and wave angles approaching the barrier can exploit local topographic lows and thin sections of the dune system, leading to an increased probability of overwash and breaching. Once breached the wave and surge angle can increase flooding landward of the barrier (Novak et al., 2024; Sánchez-Arcilla and Jiménez, 1994; Houser et al., 2008).

Original: 282 - 285
Revised: 297 - 307

Figs. 8, 9 , and 10 highlight the key findings from our simulations. Total breach area is a strong predictor of total inundation; however, breach location is also crucial, especially given the storm's forcing dynamics and surge direction. Similarly, a study by Gharagozlou et al. (2021) on breaching's impact on lagoon circulation during Hurricane Isabel illustrates how breaches alter flow patterns and introduce larger volumes of ocean water into the lagoon. These findings can be compared to our results, which demonstrate that breach location significantly influences storm surge behavior and its subsequent effects on coastal flooding (Gharagozlou et al., 2021). While this study does not include tides and waves, they significantly influence bay surge dynamics and contribute strongly to breach initiation and growth as described in  Smallegan and Irish (2017); Sherwood et al. (2014); Safak et al. (2016); Sánchez-Arcilla and Jiménez (1994). The stochastic nature to these processes makes them difficult to model, and much of our understanding relies on empirical observations from geological studies or post-storm surveys of barrier island systems  Kraus et al., 2002; Buynevich and Donnelly, 2006; Soria et al., 2021; Novak et al., 2024; Sánchez-Arcilla and Jiménez (1994). Incorporating tidal or wave components into our simulations could result in different patterns of breaching and inundation. Our use of offshore water levels to model breaching assumes wave action contributes to breach initiation, based on prior studies and observations.

**2    Conclusions**

The breaching of a barrier island during a hurricane shows a strong impact on mainland inundation. The number, locations, and size of the breaches can significantly alter the inundation pattern along the mainland coastline. While the impact of barrier-island breach during storms is unquestionable, more research is needed to better quantify the uncertainties of the breaching process. In particular, the statistical distribution of the breach parameters might vary for different barrier island systems. This is particularly important for barrier island systems that lack extensive datasets on past storms and breaching events, or for storm conditions that have yet to be observed. In this context, our work is categorized as preliminary and highlights the importance of understanding how barrier-island breaching affects the vulnerability of mainland coastal areas to storm impacts. The global prevalence of coastal barrier systems presents numerous opportunities to advance our understanding of breaching's impacts across diverse coastlines and storm conditions. This understanding offers opportunities to enhance infrastructure resilience, reduce potential loss of life, and minimize community disruptions caused by storms.

Original: 335 - 356
Revised: 354 - 377

**References**

Buynevich, I. and Donnelly, J. (2006). Geological signatures of barrier breaching and overwash, southern massachusetts, usa. *Journal of Coastal Research*, pages 112–116.

Gharagozlou, A., Dietrich, J. C., Massey, T. C., Anderson, D. L., Gorski, J. F., and Overton, M. F. (2021). Formation of a barrier island breach and its contributions to lagoonal circulation. *Estuarine, Coastal and Shelf Science*, 262:107593.

Houser, C., Hapke, C., and Hamilton, S. (2008). Controls on coastal dune morphology, shoreline erosion and barrier island response to extreme storms. *Geomorphology*, 100(3):223–240.

Irish, J. L., Frey, A. E., Rosati, J. D., Olivera, F., Dunkin, L. M., Kaihatu, J. M., Ferreira, C. M., and Edge, B. L. (2010). Potential implications of global warming and barrier island degradation on future hurricane inundation, property damages, and population impacted. *Ocean & Coastal Management*, 53(10):645–657.

Kraus, N. C. (2003). Analytical model of incipient breaching of coastal barriers. *Coastal Engineering Journal*, 45(04):511–531.

Kraus, N. C., Militello, A., and Todoroff, G. (2002). Barrier breaching processes and barrier spit breach, stone lagoon, california. *Shore & Beach*, 70(4):21–28.

Kraus, N. C. and Wamsley, T. V. (2003). Coastal barrier breaching, part 1: Overview of breaching processes.

Mandli, K. T., Ahmadia, A. J., Berger, M., Calhoun, D., George, D. L., Hadjimichael, Y., Ketcheson, D. I., Lemoine, G. I., and LeVeque, R. J. (2016). Clawpack: building an open source ecosystem for solving hyperbolic pdes. *PeerJ Computer Science*, 2:e68.

National Hurricane Center (2006). Hurricane Preparedness - Hazards.

Novak, L. P., da Rocha, T. B., Fernandez, G. B., de Oliveira Filho, S. R., de Mello Filho, M. E. T., and Pereira, T. G. (2024). Regional assessment of overwash processes in a retrograding sand barrier (paraíba do sul river deltaic complex, rio de janeiro, brazil). *Regional Studies in Marine Science*, 78:103733.

Oertel, G. F. (1985). The barrier island system.

Pierce, J. (1970). Tidal inlets and washover fans. *The Journal of Geology*, 78(2):230–234.

Roelvink, D., Reniers, A., Van Dongeren, A., De Vries, J. V. T., McCall, R., and Lescinski, J. (2009). Modelling storm impacts on beaches, dunes and barrier islands. *Coastal engineering*, 56(11-12):1133–1152.

Safak, I., Warner, J. C., and List, J. H. (2016). Barrier island breach evolution: Along-shore transport and bay-ocean pressure gradient interactions. *Journal of Geophysical Research: Oceans*, 121(12):8720–8730.

Sánchez-Arcilla, A. and Jiménez, J. A. (1994). Breaching in a wave-dominated barrier spit: The trabucador bar (north-eastern spanish coast). *Earth Surface Processes and Landforms*, 19(6):483–498.

Schmeltz, E., Sorensen, R., McCarthy, M., and Nersesian, G. (1982). Breach/inlet inter-action at moriches inlet. In *Coastal Engineering 1982*, pages 1062–1077.

Sherwood, C. R., Long, J. W., Dickhudt, P. J., Dalyander, P. S., Thompson, D. M., and Plant, N. G. (2014). Inundation of a barrier island (chandeleur islands, louisiana, usa) during a hurricane: Observed water-level gradients and modeled seaward sand transport. *Journal of Geophysical Research: Earth Surface*, 119(7):1498–1515.

Smallegan, S. M. and Irish, J. L. (2017). Barrier island morphological change by bay-side storm surge. *Journal of Waterway, Port, Coastal, and Ocean Engineering*, 143(5):04017025.

Stutz, M. L. and Pilkey, O. H. (2001). A review of global barrier island distribution. *Journal of Coastal Research*, pages 15–22.

van der Lugt, M. A., Quataert, E., van Dongeren, A., van Ormondt, M., and Sherwood, C. R. (2019). Morphodynamic modeling of the response of two barrier islands to atlantic hurricane forcing. *Estuarine, Coastal and Shelf Science*, 229:106404.

Visser, P. J. (1999). Breach erosion in sand-dikes. In *Coastal Engineering 1998*, pages 3516–3528.

Wamsley, T. V. and Kraus, N. C. (2005). Coastal barrier island breaching, part 2: Mechanical breaching and breach closure. *Coastal and Hydraulic Laboratory Techni-cal Note ERDC/CHL CHETN IV-65, US Army Engineer Research and Development Center, Vicksburg, MS*, 21.

Zhang, K. and Leatherman, S. (2011). Barrier island population along the us atlantic and gulf coasts. *Journal of Coastal Research*, 27(2):356–363.